# The Prevalence of Burnout and Its Associations with Psychosocial Work Environment among Kaunas Region (Lithuania) Hospitals’ Physicians

**DOI:** 10.3390/ijerph17103739

**Published:** 2020-05-25

**Authors:** Rasa Žutautienė, Ričardas Radišauskas, Gintare Kaliniene, Ruta Ustinaviciene

**Affiliations:** 1Department of Environmental and Occupational Medicine, Public Health Faculty, Lithuanian University of Health Sciences, Kaunas 47181, Lithuania; ricardas.radisauskas@lsmuni.lt (R.R.); gintare.kaliniene@lsmuni.lt (G.K.); ruta.ustinaviciene@lsmuni.lt (R.U.); 2Health Research Institute, Faculty of Public Health, Lithuanian University of Health Sciences, Kaunas 47181, Lithuania

**Keywords:** occupational stress, burnout, psychosocial risk, prevalence, physician

## Abstract

The primary prevention of occupational burnout should be considered as a public health priority worldwide. The aim of this study was to evaluate the prevalence of burnout and its associations with the work environment among hospital physicians in the Kaunas region, Lithuania. The cross-sectional study was carried out in 2018. The Job Content Questionnaire (JCQ) and the Copenhagen Burnout Inventory (CBI) were administered to examine occupational stress and personal, work-related, and client-related burnout among 647 physicians. Logistic regression analysis was applied to determine the association between dependent variable burnout and psychosocial environment among physicians, adjusting for potential confounders of age and gender. The prevalence rate of client-related, work-related, and personal burnout was 35.1%, 46.7%, and 44.8%, respectively. High job control, lack of supervisor, coworker support, job demands, and job insecurity were significantly associated with all three sub-dimensions of burnout. High job demands increased the probability of all three burnout dimensions, high job control reduced the probability of work-related, and client-related burnout and high job insecurity increased the probability of client-related burnout. The confirmed associations suggest that optimization of job demands and job control and the improvement of job security would be effective preventive measures in reducing occupational burnout among physicians.

## 1. Introduction

Burnout has reached epidemic levels in healthcare. Among the European medical specialties, burnout rates are between 25% and 60% [1]; therefore, health at work has become a major concern in our society and a priority in public health [2].

Chronic stress associated with emotionally intense work demands for which resources are inadequate can result in burnout [3,4]. Physicians are particularly vulnerable to experiencing burnout due to heavy workloads and high levels of work-related stresses [5]. Some studies have found that physicians were significantly more likely to experience symptoms of burnout than any other occupational group [6].

Burnout is a state of physical, emotional, and mental exhaustion that results from long-term involvement in work situations that are emotionally demanding [7]. Some studies have found that physicians’ burnout is associated with increased medical errors, suicide, cardiovascular diseases, lower patient satisfaction, longer postdischarge recovery times, and decreased professional work effort [8,9,10]. A poor psychosocial work environment could have negative effects on health, work ability, and productivity [11]. Burnout is etiologically, clinically, and nosologically similar to depression [12]. Research confirms that work environments, such as workload, night work, work experience, loss of autonomy, and lack of time to socialize with colleagues, are also significant for the development of burnout syndrome [13,14]. Burnout is also known to be related to individual characteristics, such as gender, age, and marital status [14,15,16,17,18]. The primary prevention of occupational burnout should be considered as a public health priority worldwide. There are many scientific publications examining the working environment of nurses, but there is still a lack of research on the psychosocial work environment of doctors, risk factors, and their impact on doctors’ health and quality of work in Lithuania. One valuable investigation among Lithuanian anesthetists’ and intensive care professionals’ have shown high rates of burnout that were strongly related to high workloads and low salaries [19,20], also family physicians’ psychological distress was associated with workplace bullying [21]. Therefore, more widely on the national level, this area should be investigated by including various sectors of health care. This investigation aims to reveal hospital physicians’ situations in six hospitals in the Kaunas region.

The aim of this study was to evaluate the prevalence of burnout and its associations with the work environment among hospital physicians in the Kaunas region, Lithuania.

## 2. Materials and Methods

The study design was a cross-sectional epidemiologic study that was carried out in 2018. The participation of physicians in the research was voluntary and anonymous. The study population (N = 2353) included all physicians working in 6 hospitals in the Kaunas region. The sample size calculation was based on the frequency with 5% probability of error and 95% reliability, and 0.5 relative frequency [22], and this resulted in 330 participants needed to complete the study. A total of 830 questionnaires were distributed among physicians, and 647 respondents agreed to participate in the study and completed the questionnaire properly (response rate of 81.3%). The study protocol was approved by the Kaunas Regional Ethics Committee for Biomedical Research (Protocol No. BE-2-41).

### 2.1. Questionnaires

A three-part questionnaire was used in this study. The first part of the questions revealed the demographic characteristics of the respondents (gender, age, marital status, and length of employment).

Most studies that investigate burnout use Maslach Burnout Inventory (MBI), but we chose the Copenhagen Burnout Inventory (CBI). Therefore, the second part of the questionnaire was the Copenhagen Burnout Inventory (CBI) [7]. In our study, burnout was the dependent variable. This instrument includes three domains of burnout: personal burnout (6 items), work-related burnout (7 items), and client-related burnout (6 items). All questions have 5 possible answers. Following the authors’ instruction, the answers were converted into a scoring system from 0 to 100 (always—100; often—75; sometimes—50; seldom—25; never/almost never—0). The score for each scale corresponded to the calculated mean of the scale scores. The calculated scores of scales indicate the presence of burnout if it amounts to higher than 50 points.

The third part of the instrument was a standardized Job Content Questionnaire (JCQ) [23] that had been designed to measure work environment characteristics based on the demand–control–support model. JCQ is a well-established and widely used self-report instrument that measures the work dimensions. The questionnaire comprises five scales: job demand (5 items); job control (9 items—the sum of two subscales: skill discretion measured by 6 items and decision authority measured by 3 items); supervisor support (4 items) and coworker support (4 items); and job insecurity (3 items). The questions are rated on a four-point Likert-type scale ranging from 1 (strongly disagree) to 4 (strongly agree) except job insecurity scale’s questions with different possible answers that are rated on a five-point scale. The special calculations according to the JCQ user’s instruction [23] were performed to create the final scale’s scores. High job demand and job insecurity scores and low scores for the other subscales were considered to indicate a more negative psychosocial environment in this regard.

### 2.2. Statistical Analysis

Statistical data analysis was performed using the IBM SPSS 20.0 software package (IBM Inc., Armonk, New York, NY, USA). Descriptive data were expressed as a percentage. We calculated mean scores, standard deviation (SD), median, and min/max for each of the three scales of burnout (for the personal, work-related, client-related burnout), and five scales for JCQ for the whole sample.

Hypotheses about the equality between the averages of two quantitative variables were examined by U-test (nonparametric Mann–Whitney). The difference was considered to be statistically significant when *p* < 0.05.

Spearman’s correlation was computed to estimate the direct or indirect association between three burnout dimensions and age, job control, job demand, supervisor support, coworker support, and job insecurity.

In order to determine whether the work-related factors were associated with burnout dimension, logistic regression models were applied. As a first step, univariate analysis (adjusted only by age) was performed. Controlling potentially confounding factors (age and gender). As a final step, the model of multivariate logistic regression was applied to determine the statistical significance of the association between dependent variable burnout (personal, work-related, and client-related) and the independent variables job demand, job control, supervisor support, coworker support, and job insecurity. The results are presented as regression coefficients (B), odds ratios (ORs), its 95% confidence intervals (CIs), and *p*-value. A *p*-value of < 0.05 was considered statistically significant. The accuracy and feasibility of multivariate logistic regression models were evaluated using the Classification table and Nagelkerke R^2^ test ($R_{N}^{2}$).

## 3. Results

The main socio-demographic and occupation-related individual characteristics of participants are shown in Table 1. The bigger part of the study participants was female. The mean age of the study sample was 39.7 (*SD* = 13.58). Participants had been working at their current job for an average of 14 years (*SD* = 13.19). The main proportion consisted of respondents living in partnership (married), more than a third were single, others—divorced and widowed. Fifty-two percent of respondents were therapeutic profile specialists, quarter surgical and other attributed to other specialties.

In Table 2, the mean, median, and standard deviation values of the JCQ and CBI scales are presented. The results revealed (Table 2) sufficient internal consistency of measured dimensions—almost all scales had Cronbach’s α coefficients of >0.6 [7]. Almost half of the physicians (44.8%) were classified as high personal burnout perceived group, 46.7% as high work-related burnout perceived group, and 35.1% as high client-related burnout perceived group (when the cut-off point of CBI scales scores was 50 points).

Spearman’s correlation between JCQ scales and three CBI scales are presented in Table 3. Significant direct correlations were observed between all types of burnout and job demands as well as job insecurity. Additionally, all burnout scales were in significant inverse correlation with age, length of employment (except client-related burnout), and supervisor support. However, personal and work-related burnouts were in significant inverse correlation with job control, and personal and client-related burnouts were in significant inverse correlation with coworker support. In order to investigate whether burnout differs among gender groups, we explored its differences in CBI scales. The results of our study showed that women perceived significantly higher personal ($\Sigma_{r}$ = 143,167.0) and client-related burnout ($\Sigma_{r}$ = 142,681.0) in comparison with men (respectively, $\Sigma_{r}$ = 66,461.0 and $\Sigma_{r}$ = 66,947.0). Work-related burnout average scores did not differ between gender groups ($\Sigma_{r}$ = 138,389.5 (for female) $\Sigma_{r}$ = 138,389.5 (for male)). The Mann–Whitney test for personal burnout was U = 41,708.0, *p* = 0.015; for client-related burnout, U = 42,194.0, *p* = 0.027; and for work-related burnout, U = 46,485.5, *p* = 0.759).

Further analysis of associations between burnout (CBI scales) and psychosocial factors (JCQ scales) was performed. The first step was univariate logistic regression analysis shown in Table 4.

In the final multivariate logistic regression models for burnout scales shown in Table 5, only those independent variables were included that previously were found as significant in the univariate logistic regression models. These results revealed that older age significantly reduced personal and work-related burnout probability (OR = 0.977 and OR = 0.964, respectively). Gender was found as a significant factor only for personal burnout—in this case, women had a greater probability of burnout (OR = 1.649). High job demands increased the probability of all burnout dimensions (for personal burnout OR = 2.118, for work-related burnout OR = 3.482; for patient-related burnout OR = 4.894). Physicians with high job control had significantly on average 74% and 52% lower probability of work-related and patients-related burnout, respectively (OR = 0.264 and OR = 0.479) compared with physicians with less job control. High job insecurity increased the probability of client-related burnout: physicians with a high level of job insecurity had almost two times higher probability of patients-related burnout in comparison with physicians with low job insecurity (OR = 1.973).

## 4. Discussion

The aim of this study was to evaluate the prevalence of burnout and its associations with work psychosocial environment among hospital physicians in the Kaunas region, Lithuania. Kaunas Clinics (Kauno Klinikos) is the biggest hospital in Lithuania. We examined the relationship between work environment and burnout among hospital physicians. Our results suggest that burnout is highly prevalent in this population. Very similar data were found by other researchers in Lithuania [19,20,21], but these studies cover small and very special physician population groups (anesthetists, intensive care physicians, cardiac surgeons). Burnout syndrome is common in all groups of workers (lawyers, physicians, nurses, teachers, bus drivers, and people working in advertising and information technology), but most often among physicians [21,24,25]. In this study burnout was evaluated in three sub-dimensions: personal (44.8%), work-related (46.7%), and client-related (35.1%). Burnout affects doctors, patients, and the healthcare system as a whole. The Burnout Study Group, a European network of general practice research, conducted a study in 12 European countries (Poland, Sweden, England (UK), Bulgaria, Croatia, France, Greece, Italy, Hungary, Turkey, Malta, and Spain) to determine burnout at work. The results revealed that 43% of respondents had high burnout rates [17]. The prevalence of burnout was also high (more than 40%) in Germany [26,27,28]. Very similar results were obtained in France [29] and Serbia [30]. Compared with previous studies, burnout rates that we obtained for the medical workforce were higher, ranging between 25% and 60% depending on the medical specialty [1]. A very special situation is in the Netherlands, where the researchers found the lowest prevalence of burnout in Europe. According to the study authors, burnout is found only in 4% of intensive care physicians. The authors regard such a low prevalence of burnout as an achievement of the Dutch health system, where the workload of physicians is significantly lower than in hospitals in other countries [31]. At the same time, in the Netherlands, there is a high-quality burnout treatment system and social insurance; the employer is obliged to pay salaries during the first two years of illness. Burnout is recognized as an occupational disease. Most employers are insured against this condition therefore, an insurance company takes over salary payment in case of long-term illness—such as burnout. The highest burnout level (76%) was determined among residents at Washington University Hospital. The authors did not attempt to find an association between burnout and workload or any other explanatory factors [32]. Research on burnout among physicians has increased awareness of mental health and well-being as an important issue, and US national organizations have recently invited all health care systems to assess their physicians on the measure of well-being, often with a focus on burnout [33]. In our study, a total of 36.5% of respondents (*n* = 236) had a high score of burnouts (more than 50 points). This may be related to working conditions in Lithuania: heavy workloads, low salaries in the health care sector, personal responsibility for health, and not fully functioning medical error insurance system.

Job satisfaction often depends not only on various external factors but also on personal characteristics. In our study the main causes of burnout were job demands, job insecurity, lack of supervisor, and coworker supports. Female physicians had a greater probability of burnout. High job demands increased the probability of all burnout dimensions. Physicians with high job control had a significantly lower probability of work-related and patient-related burnout, respectively. Additionally, high job insecurity increased the probability of client-related burnout. Researchers of Serbia found that despite high demands and responsibilities at work and in spite of the low degree of their autonomy, physicians do not experience work-related burnout [30]. Very similar causes were established in France: 47% of intensive care physicians experience burnout syndrome due to job demands, conflicts with a coworker or supervisor; women were also at a greater risk of burnout compared with men [29]. In terms of gender differences in burnout, the results are inconsistent. However, our results add knowledge to the investigations, which confirm that women are more prone to high perceived burnout. Literature shows evidence that burnout among female physicians significantly correlates with a higher number of children [34]. These findings suggest that improving work-related factors with targeted interventions, including a supportive work environment, may increase life satisfaction among doctors [35]. The most commonly studied interventions have involved mindfulness, stress management, and small group discussions. The results suggest that these strategies can be effective approaches to reduce burnout domain scores [3]. Health promotion measures that reduce the risk of burnout for healthcare providers can also help improve the quality of healthcare. On the other hand, interventions must differ according to the country, legislation system, and interpersonal traditions.

For our study, we chose the Copenhagen Burnout Inventory (CBI). Studies provided support for its validity and the instrument was used in previous studies with healthcare providers [26]. Compared with the more frequently used multidimensional Maslach Burnout Inventory (MBI), personal burnout according to the CBI can be measured in one total score. A study using the MBI and the CBI among dentists in Australia concludes that there are similarities between the two measures, and that the CBI has better psychometric properties and is an appropriate measure of burnout among healthcare professionals [26]. This indicates that the CBI scales do not measure the stable traits of individuals, but degrees of burnout that may change over time [7]. Another instrument used in this study was JCQ. The Job Content Questionnaire (JCQ) is a well-established and widely used self-report instrument that measures the work dimensions based on the DC/DCS (Job Demands–Control/Demand–Control–Support) model in the workplace [36]. For over 40 years, occupational health studies using the JCQ have shown evidence that high levels of job strain have negative effects on many health outcomes, including cardiovascular, musculoskeletal, and psychiatric diseases [37] in various working populations, including medical settings [37].

### Limitations

Several methodological limitations of our research need to be considered. It is a cross-sectional study, and all data are collected on physicians’ own reports, so data can include biases of self-reporting. On the other hand, the population of physicians is highly educated and, therefore, answers difficult questions well and correctly. The response rate in our study was high, but it is possible that those physicians who chose not to respond already suffered from burnout, so our data might be underestimated. That is, we might have obtained data about even higher levels of burnout. Additionally, it must be indicated that some important information such as physicians’ specialties and some socioeconomic conditions (income, living conditions) were not included in this manuscript.

Despite the limitations, our study also has advantages and is significant from a practical point of view. First of all, this is one of the extensive studies aimed at finding out the correlations between work environment factors and burnout among Lithuanian physicians. In Lithuania, there is a shortage of doctors in almost all specialties, a large emigration of doctors, a great shortage of doctors in rural areas, low salaries, and a great shortage of middle-level staff.

The study used standardized instruments frequently used in this type of research: The Job Content Questionnaire was used to measure work environment characteristics based on the demand–control–support model, and the Copenhagen Burnout Inventory was used for evaluation of burnout at work. The study involved physicians of various specialties and interviewed a large number of physicians. In addition, our study is one of the biggest in Lithuania that examines the prevalence of burnout among physicians and identifies areas that need to be examined in detail in the future. The problems we have identified may be relevant to all Eastern European countries with similar health systems and which have undergone significant changes in recent decades.

## 5. Conclusions

The results of the current study demonstrated that the prevalence of burnout was high among Kaunas region hospital physicians. The significant associations were found between job demands, job control, job insecurity, and supervisor support, and burnout among physicians. However, the final analysis model revealed that the most significant predictor for all burnout dimensions was job demands. Job control was a significant predictor for work-related burnout and client-related burnout, job insecurity for client-related burnout. The future attention in creating new effective interventions for reducing occupational burnout should be paid to these psychosocial work environment aspects—job demands, job control, and job insecurity.

## Figures and Tables

**Table 1 ijerph-17-03739-t001:** Individual characteristics of the study population.

Characteristics	n (%)
**Gender**	
Men	222 (34.3)
Women	425 (65.7)
**Marital Status**	
Single	240 (37.1)
Married	332 (51.3)
Divorced	56 (8.7)
Widowed	19 (2.9)
**Specialties**	
Surgical	163 (25.2)
Therapeutic	340 (52.6)
Other (not specified)	144 (22.2)

**Table 2 ijerph-17-03739-t002:** Baseline characteristics (in scores) of the psychosocial work environment (occupational stress dimensions) by Job Content and Copenhagen Burnout Inventory Questionnaires.

Occupational Stress Dimensions	Cronbach α	Mean (SD)	Median	Minimum/Maximum
**JCQ**				
Job control	0.75	70.93 (10.30)	72.00	32/96
Job demand	0.56	33.2 (4.81)	33.0	18/48
Supervisory support	0.89	11.53 (2.37)	12.00	4/18
Coworker support	0.75	11.98 (1.63)	12.00	4/17
Job insecurity	0.54	5.38 (1.63)	5.00	3/12
**CBI**				
Personal burnout	0.84	45.27 (17.77)	45.83	0/100
Work-related burnout	0.83	46.41 (17.16)	46.43	0/96.4
Client-related burnout	0.83	40.16 (17.97)	41.67	0/100

Note: SD—standard deviation.

**Table 3 ijerph-17-03739-t003:** Association between Job Content and age, length of employment, and Copenhagen Burnout Inventory Questionnaires scores by Spearman correlation analysis.

	Personal Burnout	*p*	Work-Related Burnout	*p*	Client-Related Burnout	*p*
**Age**	−0.180	<0.001	−0.248	<0.001	−0.029	0.461
**Length of employment**	−0.174	<0.001	−0.223	<0.001	−0.022	0.571
**Job control**	−0.155	<0.001	−0.261	<0.001	−0.075	0.056
**Job demand**	0.148	<0.001	0.202	<0.001	0.326	<0.001
**Supervisor support**	−0.138	<0.001	−0.195	<0.001	−0.201	<0.001
**Coworker support**	−0.049	0.213	−0.038	0.333	−0.176	<0.001
**Job insecurity**	0.135	0.001	0.123	0.002	0.307	<0.001

**Table 4 ijerph-17-03739-t004:** Factors associated with burnout among physicians (univariate logistic regression analysis).

Factor	Personal Burnout *			
	B	OR	95% CI	*p*
**Age**	−0.023	0.977	0.966–0.989	<0.001
**Gender**	0.503	1.654	1.181–2.316	0.003
**Length of employment**	0.008	1.008	0.962–1.056	0.732
**Job control**	0.020	1.020	0.598–1.740	0.942
**Job demand**	0.755	2.128	1.255–3.609	0.005
**Supervisor support**	−0.957	1.002	0.403–1.623	0.852
**Coworker support**	0.463	1.589	0.593–4.261	0.357
**Job insecurity**	0.405	1.499	0.981–2.291	0.062
	**Work-Related Burnout ***			
	B	OR	95% CI	*p*
**Age**	−0.038	0.963	0.951–0.975	<0.001
**Gender**	−0.054	0.948	0.678–1.325	0.754
**Length of employment**	0.049	1.051	0.996–1.108	0.071
**Job control**	−1.124	0.325	0.174–0.609	<0.001
**Job demand**	1.056	2.875	1.639–5.043	<0.001
**Supervisor support**	−1.012	0.295	0.102–1.983	0.635
**Coworker support**	0.582	1.789	0.642–4.983	0.266
**Job insecurity**	0.407	1.503	0.974–2.318	0.065
	**Client-Related Burnout ***			
	B	OR	95% CI	*p*
**Age**	−0.003	0.997	0.985–1.009	0.657
**Gender**	0.060	1.062	0.755–1.494	0.729
**Length of employment**	−0.005	0.995	0.949–1.043	0.821
**Job control**	−0.477	0.621	0.342–0.925	0.016
**Job demand**	1.545	4.687	2.712–8.102	<0.001
**Supervisor support**	−1.101	0.105	0.013–2.956	0.759
**Coworker support**	−1.447	0.235	0.053–1.040	0.056
**Job insecurity**	0.757	2.131	1.397–3.253	<0.001

Note: B—regression coefficients; OR—odds ratios; 95% CI—95% confidence intervals; *—date adjusted by age.

**Table 5 ijerph-17-03739-t005:** Factors associated with burnout among physicians (multivariate logistic regression analysis).

Factor	Personal Burnout *			
	B	OR	95% CI	*p*
Age	−0.024	0.977	0.965–0.989	<0.001
Gender (female)	0.5000	1.649	1.175–2.313	0.004
Job demand (high scores)	0.750	2.118	1.244–3.604	0.006
Classification Table 56.6%; Nagelkerke R Square 0.064				
	**Work-Related Burnout ***			
	B	OR	95% CI	*p*
Age	−0.036	0.964	0.952–0.976	<0.001
Job control (high scores)	−1.330	0.264	0.137–0.511	<0.001
Job demand (high scores)	1.248	3.482	1.927-6.292	<0.001
Classification Table 62.1%; Nagelkerke R Square 0.141				
	**Client-Related Burnout ***			
	B	OR	95% CI	*p*
Job control (high scores)	−0.737	0.479	0.250–0.915	0.026
Job demand (high scores)	1.588	4.894	2.790–8.583	<0.001
Job insecurity (high scores)	0.680	1.973	1.278–3.046	0.002
Classification Table 68.6%; Nagelkerke R Square 0.099				

Note: B—regression coefficients; OR—odds ratios; 95% CI—95% Confidence interval; *—data adjusted by gender, age.

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
