# Peer review of "The Prevalence of Burnout and Its Associations with Psychosocial Work Environment among Kaunas Region (Lithuania) Hospitals’ Physicians"

_ijerph, 2020, doi:10.3390/ijerph17103739_

Round 1

Reviewer 1 Report

Thank you for the opportunity to review your article. The study aims to assess the measurement of burnout in physicians in Kaunas region (Lithuania) through the use of specific scales. The study is very interesting, well conducted and add new information for the readers.The study has clear objectives and a good methodology. The results and conclusion are consistent with the data collected.

 I suggest some integrations explained in the following specific comments:

  • Line 27: Change occupation with occupational
  • Line 41: the description of the selected existing scales for the measurement of burnout and the motivation of the choice of the ones used in the study should be moved to the Method section. Then, the authors should move the lines 40-42 to the beginning part of "Questionnaires section" in Materials and Methods
  • Line 55: the authors should cite some references about the situation in Lithuania
  • Line 100: Change 52% with Fifty two percent
  • Line 101: Remove the hyphen after 25.6%
  • Line 183: Remove a space before the word "Female"
  • Line 198: change to reducing with to reduce

Author Response

Response to Reviewer 1 Comments

Thank You very much for valuable review. We took into account your valuable suggestions and improved these points:

    Line 27: Change occupation with occupational

Response: It was changed according suggestion; occupation was changed in to occupational.

    Line 41: the description of the selected existing scales for the measurement of burnout and the motivation of the choice of the ones used in the study should be moved to the Method section. Then, the authors should move the lines 40-42 to the beginning part of "Questionnaires section" in Materials and Methods

Response: These sentences from Introduction section were moved to Materials and Methods section (lines: 82, 86, 93, 94).    

    Line 55: the authors should cite some references about the situation in Lithuania

Response: More accurate situation among Lithuanian physicians based on scientific publications was presented (lines 56-61).

    Line 100: Change 52% with Fifty two percent

Response: It was changed according suggestion; changed 52% with Fifty two percent.

    Line 101: Remove the hyphen after 25.6%

Response: It was changed according suggestion; removed the hyphen after 25.6%.

    Line 183: Remove a space before the word "Female"

Response: It was changed according suggestion; removed a space before the word “female”.

    Line 198: change to reducing with to reduce

Response: It was changed according suggestion; changed to reducing with to reduce.

Reviewer 2 Report

Zutautiene et al.: The prevalence of burnout and its associations with psychosocial work environment among Kaunas region (Lithuania) hospitals’ physicians

This is a cross-sectional epidemiological study aimed to survey level of burnout among hospital physicians in Lithuania. Altogether 647 persons participated in the survey. High rates of client-related, work-related and personal burnout were detected. Burnout is an increasing problem in health workers, thus the topic is worth for investigation.

Detailed comments:

Introduction, line 40: next 3 sentences should go to the Materials and Methods section. At the same time please transfer the explanation why Copenhagen Burnout Inventory (CBI) and Job Content Questionnaire (JCQ) was chose from the last paragraph of Discussion to the Materials and Methods section.  

To give an impression about the situation of doctors in Lithuania for the international readers, authors should insert a brief paragraph about this issue into the Introduction.

Materials and Methods, Questionnaires: authors transformed continuous variables (age, length of employment) into categorical variables and argued that ‘It was done for the purpose of more accurate statistical analyses’. I suggest using appropriate statistical methods for the analyses instead of transformation.

Materials and Methods: as burnout is depending on the specialty of the physician, distribution of the participants’ specialty would be important information.

Materials and Methods section: statistical methods are missing.

Results, 1st paragraph: it is not necessary repeating the results in the text that has already been shown in Table 1.

Results, line 104: authors state here that ‘results revealed high internal consistency of measured dimensions - almost all scales had Cronbach’s α coefficients >0.6. Usually a Cronbach’s α above 0.8 is considered high. ’Job demand and job insecurity have a definitely low Cronbach’s coefficients. Please correct this sentence!

Results, Table 3: Repeating the results of the Table in the text is not necessary. The title of the Table is misleading as the Table contains the correlations of age too.

Results, line 131: authors mention here that univariate logistic regressions was used for the analyses and only those independent variables were included that previously showed significant correlation. The results of the correlation analysis should be given.

Author Response

Response to Reviewer 2 Comments

Thank You very much for valuable review. We took into account your valuable suggestions and improved these points:

Introduction, line 40: next 3 sentences should go to the Materials and Methods section. At the same time please transfer the explanation why Copenhagen Burnout Inventory (CBI) and Job Content Questionnaire (JCQ) was chose from the last paragraph of Discussion to the Materials and Methods section. 

Response: It was changed according suggestion (lines: 82, 86, 93, 94).

To give an impression about the situation of doctors in Lithuania for the international readers, authors should insert a brief paragraph about this issue into the Introduction.

Response: More accurate situation among Lithuanian physicians based on scientific publications was presented (lines 56-61).

Materials and Methods, Questionnaires: authors transformed continuous variables (age, length of employment) into categorical variables and argued that ‘It was done for the purpose of more accurate statistical analyses’. I suggest using appropriate statistical methods for the analyses instead of transformation.

Response: The categorized variables were presented only in descriptive statistics (Table 1), in relationships’ analysis – in correlation and logistic regression analysis we used continuous variables of age and length of employment. It seems that transformation was not necessary, therefore the data showing categorised variables were removed (Table 1).

Materials and Methods: as burnout is depending on the specialty of the physician, distribution of the participants’ specialty would be important information.

Response: it was not the purpose of this study to analyse the impact of physicians’ specialities but the detailed comparison between specialities are the object of further study of this physicians’ population. Therefore this aspect we included in to Limitations’ section of this manuscript (lines: 260-262).

Materials and Methods section: statistical methods are missing.

Response: This part was not included by our mistake. It was improved in revised version of manuscript (lines: 104-125). 

Results, 1st paragraph: it is not necessary repeating the results in the text that has already been shown in Table 1.

Response: It was changed according suggestion.

Results, line 104: authors state here that ‘results revealed high internal consistency of measured dimensions - almost all scales had Cronbach’s α coefficients >0.6. Usually a Cronbach’s α above 0.8 is considered high. ’Job demand and job insecurity have a definitely low Cronbach’s coefficients. Please correct this sentence!

Response: It was changed according suggestion (line 137). 

Results, Table 3: Repeating the results of the Table in the text is not necessary. The title of the Table is misleading as the Table contains the correlations of age too.

Response: It was changed according suggestion (lines: 158,159).

Results, line 131: authors mention here that univariate logistic regressions was used for the analyses and only those independent variables were included that previously showed significant correlation. The results of the correlation analysis should be given.

Response: The univariate logistic regression model was presented in Table 4, line 166.    

Reviewer 3 Report

Thank you for providing me the opportunity to review this paper reporting on the results of a survey looking at risk of burnout in hospital doctors. The paper was well prepared and the study was clearly presented. I have only few questions.

Methods

Sampling: Were there any differences (e.g. sociodemographic or workplace-related) between participants and non-participants?

Discussion

Did the survey asked about variables such as income, living conditions, quality of family relationships? Could these variables have any importance?

God luck with finalizing the paper.

Author Response

Response to Reviewer 3 Comments

Thank You very much for valuable review. We took into account your valuable suggestions and improved these points:

Sampling: Were there any differences (e.g. sociodemographic or workplace-related) between participants and non-participants?

Response: The study response rate was 81.3%. We have not any data of non–participants (18.7% of study population). We can only hypothesize that respondents who disagreed to participate in our study probably have higher prevalence of burnout.

Did the survey asked about variables such as income, living conditions, quality of family relationships? Could these variables have any importance?

Response: These variables (such as income, living conditions, quality of family relationships) were not included in our investigation. However, they probably would have some importance especially for personal burnout but we can only hypothesize about these associations. Therefore these aspect we included in to Limitations’ section of this manuscript (lines 260-262).

Round 2

Reviewer 2 Report

My comments were adequatelly addressed by the authors. Accept as it is.